# The Association between Non-Alcoholic Fatty Liver Disease and Dynapenia in Men Diagnosed with Type 2 Diabetes Mellitus

**DOI:** 10.3390/healthcare11020243

**Published:** 2023-01-13

**Authors:** Atilla Bulur, Rıdvan Sivritepe

**Affiliations:** 1Gastroenterology and Endoscopy Private Clinic, Aydin 09800, Turkey; 2Department of Internal Medicine, Faculty of Medicine, Istanbul Medipol University, Istanbul 34810, Turkey

**Keywords:** type 2 diabetes mellitus, non-alcoholic fatty liver disease, dynapenia

## Abstract

**Background:** Dynapenia and non-alcoholic fatty liver disease (NAFLD) are common, especially in the middle and advanced-age diabetic male population. We aimed to examine the clinical features, NAFLD severity, and parameters associated with the presence of dynapenia in type 2 diabetes mellitus (T2DM) cases. **Material and Methods:** One hundred thirty-five male patients diagnosed with T2DM between 45 and 65 years of age were included. Patients were staged by ultrasonography according to NAFLD status. **Results:** There were significant differences in muscle strength, upper arm circumference, calf circumference, and up-and-go test scores between the mild-moderate-severe and non-NAFLD groups (*p* < 0.001 for all). The frequency of dynapenia was lower, and arm and calf circumferences were higher in patients without NAFLD. The muscle strength, upper arm circumference, calf circumference, and up-and-go test scores were significantly lower in the dynapenic group compared to the non-dynapenic group (*p* < 0.005 for all). The prevalence of dynapenia increased along with the increase in NAFLD stages (*p* < 0.001). **Conclusions:** We detected a significant association between NAFLD and dynapenia in middle-aged men with T2DM. As muscle strength decreases, the amount of fat in the liver increases, and as the fat in the liver increases, muscle strength decreases.

## 1. Introduction

Type 2 Diabetes Mellitus (T2DM) is one of the most common noncommunicable diseases in developing countries. Today, more than 400 million people are known to have T2DM, and this number is estimated to reach 642 million by 2040. NAFLD refers to the presence of hepatic steatosis without a condition to cause secondary hepatic fat deposition. Studies conducted in the USA reported NAFLD prevalence between 10% and 46%. In western industrialized countries where central obesity, T2DM, dyslipidemia, and metabolic syndrome are common, the most common liver disease associated with these diseases is NAFLD. Therefore, it is predicted that NAFLD prevalence will increase along with the increase in T2DM incidence [1,2,3].

Insulin resistance is one of the most important risk factors for T2DM and NAFLD. However, the mechanisms underlying the development of NAFLD and T2DM are still not fully clarified. Furthermore, hepatic fat accumulation, changes in mitochondrial function and energy metabolism, and inflammatory cytokines produced from various cell types, including immune cells, lipotoxins, and adipocytokines, have been suggested to play an important role in the development of both NAFLD and T2DM [4].

Dynapenia is a decreased muscle strength without a reduced muscle mass due to ageing. Theories have been presented that dynapenia may be caused by the disturbances in the stimulation–contraction process in the skeletal muscle along with the effect of many neuronal factors such as the deterioration in the ability of the nervous system to innervate the skeletal muscle, having fewer motor neuron units with ageing, and a decrease in the excitability of neurons [5,6,7].

Dynapenia and NAFLD are becoming increasingly common, especially in the middle and elderly diabetic male population, reducing physical performance and quality of life; The treatment process, costs, and complications constitute an important social problem [5,6,7].

In our study, we aimed to examine the clinical features, NAFLD severity, and parameters that may be associated with dynapenia in middle-aged male T2DM patients and to reveal the associated factors. Such a clinical study has yet to be completed in the literature, adding value to our study.

## 2. Materials and Methods

This study was a cross-sectional, single-center, and prospective study. All procedures performed in studies involving human participants followed the institutional and national research committee’s ethical standards and the 1964 Helsinki Declaration and its later amendments or comparable ethical standards. The study was approved by the Clinical Research Ethics Committee of the University of Health Sciences, Umraniye Training and Research Hospital (Date: 29 April 2021, No: 106). The study’s purpose and procedures were explained to all participants, and written informed consent was obtained from them. The study enrolled one hundred thirty-five male patients due to statistical power analysis. We calculated the power analysis as follows: the minimum number of patients required to complete our study, which was calculated using the descriptive statistics in the article by Zhang et al., with a confidence level of 95% (α = 0.05) and a power of 80% was 131 patients (Hintze, J. (2011). PASS 11. NCSS, LLC. Kaysville, UT, USA. www.ncss.com) (accessed on 23 September 2021) [8]. According to the admission order, male patients who were admitted to our internal medicine outpatient clinic of Istanbul Beykoz State Hospital in Turkey were included in the study. Patients between 45 and 65 years of age diagnosed with T2DM, without acute diabetic complications within the last three months, and with normal kidney and liver function tests (GFR > 90 mL, ALT < 55 U/L, AST < 34 U/L) were included in the study. The patients below 45 years and over 65 years, as well as those with T1DM, acute or chronic infection, neurological disease, history of major surgery, heavy alcohol consumption (210 gr (for men)/week), autoimmune hepatitis, acute or chronic viral hepatitis, cirrhosis, use of drugs associated with fatty liver disease and malignancy were not included in the study. Detailed anamnesis was taken from all patients, and physical examinations were performed.

Blood samples were taken on an empty stomach in the morning and analyzed simultaneously in the same laboratory. T2DM was diagnosed according to ADA criteria [9].

Anthropometric measurements such as height and body weight of the patients were measured according to standardized guidelines: Arm circumference below 22 cm was considered as ‘lower’, and at and above 22 cm, levels were considered as ‘normal’. The calf circumference measurements below 31 cm were considered ‘lower’, and at and above 31 cm were considered ‘normal’ [10].

The physical performance of the patients was evaluated by up and go test. The performance score was evaluated as follows; 1 = normal, 2 = very slightly abnormal, 3 = slightly abnormal, 4 = moderately abnormal, and 5 = severely abnormal [11].

Using a hydraulic dynamometer, the right-left hand muscle strength of the patients was measured (12-0240 Baseline Hydraulic Hand Dynamometer 200 LB Standard W/Case). Muscle strength measurements were performed before blood sampling not to affect the test. Among male patients, more than 30 were normal, and less than 30 were accepted as decreased muscle strength in the measurements [12].

Patient values such as muscle mass, body fat ratio, metabolism/body fat, and lean body mass volumes were measured using a multi-frequency bioimpedance analysis (BIA) device (A multi-frequency bioimpedance analysis (BIA) device, 2015 (InBody 720, InBody Japan Inc., Tokyo, Japan). The cut-off value for skeletal muscle mass index (SMMI) ≥ 10.76 kg/m^2^ was accepted as normal, 8.51–10.75 kg/m^2^ was accepted as moderate muscle mass reduction, and ≤8.50 was accepted as severe muscle mass reduction for men. Dynapenia was defined as a combination of normal SMMI index and decreased muscle strength, according to Manini and Clark [13]. The patients were then divided into two groups: dynapenic and non-dynapenic.

A radiologist evaluated the presence and staging of NAFLD through transabdominal ultrasonography (USG) (Acuson Sequoia 512, Siemens, Washington, DC, USA). The criteria of the Turkish Gastroenterology Association were used for the diagnosis of the fatty liver through USG. Hepatic echogenicity typically consists of measurements of the echo difference between the liver and kidneys, evaluation of echo penetration into the deep part of the liver, and determination of the clarity of blood vessel structures. The degree of lipoidosis severity was evaluated clinically as used in daily practice by dividing it into four classes. The ultrasonography findings in hepatic steatosis are presented in Figure 1 (Figure 1). Grade-0: no lipoidosis; grade-I: mild NAFLD; grade-II: moderate NAFLD; grade-III: severe NAFLD [14,15,16]. All parameters were compared between these groups, and the association between NAFLD and dynapenia in diabetic men was evaluated.

The statistical analysis was conducted through the MedCalc Statistical Software version 12.7.7 (MedCalc Software bvba, Ostend, Belgium). The normality of continuous variables was analyzed using Shapiro–Wilk’s test. Descriptive statistics were expressed by the mean and standard deviation for normally distributed variables and the median for the non-normally distributed variables. Non-parametric statistical methods were utilized to analyze the values with skewed distribution. The Mann–Whitney U test was used to compare two non-normally distributed groups. The Kruskal–Wallis test was used to compare two non-normally distributed variables. As a pairwise post hoc comparison, Bonferroni corrected Mann–Whitney U test was performed. Categorical variables were evaluated by χ^2^ test and expressed as observation counts (and percentages). Statistical significance was accepted when the two-sided *p*-value was lower than 0.05.

## 3. Results

The age average of the participants was 55.2 ± 4.6 (45–65) years, the mean muscle strength was 35.5 kg, the mean HbA1c was 9.9%, and the mean diabetes duration was 5.9 years. USG evaluation of the patients revealed that NAFLD was not detected in 34 (25.2%) patients, mild NAFLD was detected in 34 (25.2%) patients, and moderate and severe NAFLD were detected in 48 (35.6 %) and 19 (14.1%) patients, respectively. Forty-two (31.1%) patients met the criteria for dynapenia, and 93 (68.9%) patients were non-dynapenic. The demographic data, anthropometric measurements, and clinical and biochemical parameters of the patients are summarized in Table 1 (Table 1).

There were significant differences in muscle strength, upper arm circumference, calf circumference, and up-and-go test scores between the NAFLD groups (*p* < 0.001 for all). We detected that arm and calf circumferences were higher in patients without NAFLD; physical performance was better, and dynapenia prevalence was lower. There was no difference between the groups regarding other parameters, which were summarized in Table 2 and Table 3, Figure 2.

There was no lipoidosis in four patients in the dynapenic group; lipoidosis was not detected in 30 patients in the non-dynapenic group. However, the prevalence of dynapenia increased, especially in the moderate and severe NAFLD groups. The distribution of dynapenia is present in Table 4 (Table 4).

The patients were then compared according to their dynapenic status as dynapenic and non-dynapenic. All the comparisons are summarized in Table 5 (Table 5). The muscle strength, upper arm circumference, calf circumference, and up-and-go test scores were significantly lower in the dynapenic group when compared to the non-dynapenic group (*p* < 0.001, *p* < 0.001, *p* < 0.001, *p* = 0.044, respectively). There was not any statistically significant difference between the two groups for age, muscle mass, SMMI index, HbA1c, and fasting glucose measurements. The severe NAFLD ratio was 33.3% in the dynapenic group and 5.4% in the non-dynapenic group.

## 4. Discussion

In this cross-sectional study of middle-aged men with T2DM, we investigated the possible relationship between dynapenia and NAFLD, and we detected a clear connection between dynapenia and NAFLD in this patient group. We found that the muscle strength decreased along with the increase of the NAFLD stage in these patients, and the NAFLD stage increased as the muscle strength decreased. Based on the literature review, we believed that our study was valuable as it was the first study to be conducted on this subject among diabetic men. As part of our study, we included patients with T2DM who had insulin resistance, which was considered to occur similarly to NAFLD pathogenesis. Moreover, the study excluded women with different normal ranges of dynamometric measurements and younger men with no history of dynapenia. As a result, a homogeneous and specific patient group was formed. The number of patients with T2DM and NAFLD, commonly observed together, increases globally. NAFLD and T2DM occur with similar metabolic processes, and insulin resistance is considered an essential risk factor in developing both diseases [4]. The interaction between the liver, muscle, and adipose tissue, in which insulin resistance is critical, plays a vital role in developing NAFLD [16]. The skeletal muscle strongly affects glucose homeostasis and insulin resistance [17]. There is a decrease in muscle mass, muscle strength, and function in patients with T2DM. Many studies have been conducted on sarcopenia; however, studies on dynapenia are still fresh. It has been shown that dynapenia poses a risk for functional limitations and mortality in older adults. Ageing is closely associated with increased lipid accumulation in lean tissues and organs [7,18]. The liver is the main organ affected by ectopic fat accumulation. A significant association was found between increased body fat and dynapenia in patients with T2DM. It also shows that muscle weakness due to hyperglycemia may begin in the early stages of diabetes and at earlier ages. The aging skeletal muscle and chronic hyperglycemia cause the accumulation of “advanced glycation and products” in the muscle, which is thought to reduce muscle strength and muscle mass in patients with T2DM [19]. A previous study on the association between SMMI and hepatic steatosis in patients with T2DM reported that SMMI was negatively associated with hepatic steatosis in men with T2DM but not in women [20]. A previous study revealed a reverse association between hand grip strength and insulin resistance and between the glucose level at the 2nd hour in an oral glucose tolerance test [21]. It was shown that hyperglycemia assessed by HbA1c in patients with T2DM is associated with dynapenia [22]. Nishikawa et al. reported that dynapenic patients were more related to metabolic syndrome in patients with chronic liver disease than sarcopenic patients. In the same study, the frequency of dynapenia increases in patients with a fasting glucose level above 110 mg/dL. They concluded that insulin resistance might be related to muscle strength rather than muscle mass [23]. Our study consisted of patients with T2DM; unlike the studies mentioned above, there was no significant difference between HbA1c averages and dynapenia, and the dynapenia prevalence increased along with the increase in the NAFLD stage, regardless of the HbA1c value. The mean HbA1c measured was 10.4% in the group with dynapenia and 9.4% in the group without dynapenia; however, the difference was insignificant.

Sarcopenia is more common in patients with lower BMI; however, it was reported that dynapenia is more common in patients with higher BMI. It was also reported that dynapeniais is more common in patients with T2DM rather than sarcopenia as a muscle disorder form [17]. Our study found no significant difference in BMI between dynapenic and non-dynapenic groups in patients with T2DM.

NAFLD was not detected in 25.2% of our patients through USG, whereas NAFLD of various grades was found in 74.8% of patients. Furthermore, 31.1% of our patients were dynapenic, and 68.9% were non-dynapenic. When the patients were compared according to NAFLD stages, there was a significant difference between the groups in muscle measurement parameters such as muscle strength, upper arm circumference, calf circumference, and the up-and-go test. We detected that arm and calf circumferences were higher in patients without steatosis, better physical performance, and lower dynapenia prevalence. Dynapenia was detected in 45.83% of patients with moderate NAFLD and 73.68% with severe NAFLD. Moreover, a significant decrease was observed in muscle strength measurements along with the increase in NAFLD stages. In other words, it was observed in our study that the incidence of dynapenia increased along with the rise in the NAFLD stage.

Similarly, long-term progressive resistance training and muscle strengthening training has been shown to stimulate muscle hypertrophy and type II muscle fiber growth, thus preventing or limiting dynapenia [24].

In our study, no significant difference was found regarding fasting glucose, HbA1c, and BMI in all stages of NAFLD. Most guidelines include efforts to reduce oxidative stress and improve insulin resistance and weight loss for the prevention and treatment of NAFLD as management of NAFLD. However, recent studies have shown the importance of maintaining and increasing muscle mass to prevent NAFLD [25].

Our study had some limitations. Our work was conducted within a certain period. Thus, it took a short observation period to establish a causal relationship for the significant association between dynapenia and NAFLD. Our study was single-center; therefore, our results may not represent all patients with T2DM. The patients included in our study were adult, middle-aged men; consequently, it is unclear whether our findings also apply to men and women in this age group. Another limitation of the study is that the measurements and comparisons were carried out only once. Our further studies will be to observe changes in NAFLD scores with liver-protective diets and lifestyle recommendations and then to monitor changes in dynapenia scores in the same patients.

Despite these limitations, our study shows the relationship between NAFLD and dynapenia in middle-aged adult T2DM patients and is a valuable study that has not been conducted before.

## 5. Conclusions

In conclusion, the results of our study revealed that maintaining fasting glucose, HbA1c, and BMI of the patients under control will lead to the improvement of NAFLD disease developed in middle-aged male patients with T2DM along with improving muscle strength However, further studies with more extensive series should be performed to support our hypothesis.

## Figures and Tables

**Figure 1 healthcare-11-00243-f001:**
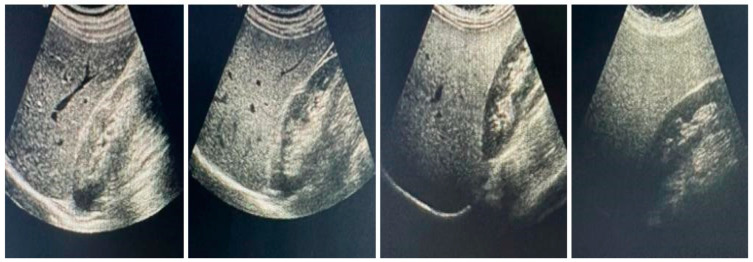
Ultrasonography findings in hepatic steatosis. Liver with no increase in echogenicity compared to kidney. From left to right; Fatty liver is excluded in the first picture, while mild, moderate and severe hepatic steatosis is seen, respectively [15].

**Figure 2 healthcare-11-00243-f002:**
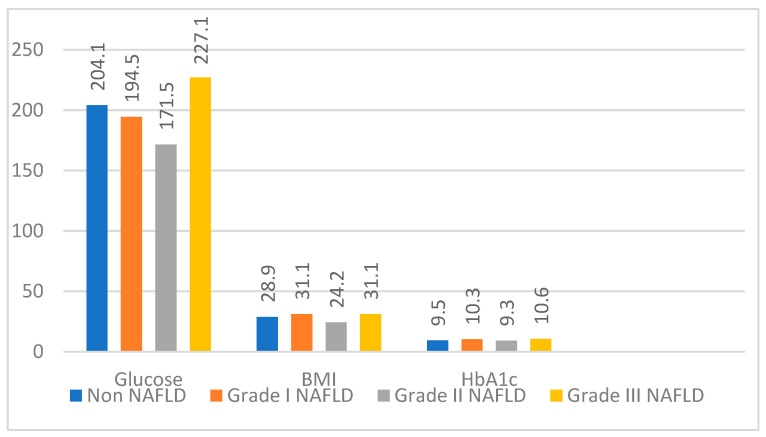
Fasting glucose (n = mg/dL), BMI (n = n) and HbA1c (n = %) measurements according to NAFLD stages *(NAFLD: Non-alcoholic fatty liver disease; BMI: Body mass index)*.

**Table 1 healthcare-11-00243-t001:** Demographic data, anthropometric measurements, clinical and biochemical parameters of the patients.

Parameter	Median	Minimum	Maximum
**Age (years)**	55	45	65
**Height (cm)**	165	145	189
**Weight (kg)**	76	44	130
**BMI (kg/m^2^)**	28.6	17	49.3
**Muscle Strength (kg)**	36	18	60
**Upper arm circumference (cm)**	24	14	41
**Calf circumference (cm)**	32	21	39
**SMMI index**	39.4	26.6	61.8
**Hba1c (4.7–5.6%)**	9.6	6.5	15.3
**Glucose (70–100 mg/dL)**	188	69	373
**Creatinine (<1 mg/dL)**	0.7	0.5	1.1
**C-reactive protein (<3 mg/L)**	9	0.2	25
**Diabetes Duration (years)**	6.1	1	23
**HDL (40–60 mg/dL)**	33	5	75
**Total Cholesterol (<200 mg/dL)**	245	107	359
**Triglyceride (<150 mg/dL)**	145	59	321
**Leukocyte (4.1–8.9 10^3^/uL)**	9	3	15.3
**Blood Urea Nitrogen (10–20 mg/dL)**	32	15	57
**Hemoglobin (12.4–14.8 g/L)**	14.3	10.8	17.7
**Platelet (150.000–450.000/mm^3^)**	176	100	485

BMI: Body mass index. SMMI index: Skeletal muscle mass index. HBA1C: Hemoglobin A1c. HDL: High-density lipoprotein cholesterol. LDL: Low-density lipoprotein cholesterol.

**Table 2 healthcare-11-00243-t002:** Review of demographic, body and muscle measurements and laboratory features of the patients according to NAFLD groups.

Med (Min-Max)	Non-NAFLD	Mild NAFLD	Moderate NAFLD	Severe NAFLD	*p* Value
**Age (years)**	54.5 (46–65)	57 (47–65)	54 (45–65)	55 (45–65)	0.626
**BMI (kg/m^2^)**	28.9 (17–49.3)	31.1 (19.6–41.3)	24.2 (18,6–43,8)	31.1 (17.4–42)	0.362
**Muscle Strength (kg)**	44 (24–50)	38 (28–60)	30 (18–50)	24 (18–42)	**<0.001 ***
**SMMI %**	39.1 (28.9–48.3)	39.4 (27.1–47)	40 (26.6–61.8)	38.4 (29.2–44)	0.312
**Upper arm circumference (cm)**	28 (15–34)	26 (24–27)	21 (17–24)	18 (14–41)	**<0.001 ***
**Calf circumference (cm)**	34 (21–39)	33.5 (28–39)	30.5 (26–36)	29 (21–39)	**<0.001 ***
**Hba1c (4.7–5.6%)**	9.5 (6.5–14.2)	10.3 (6.5–14.5)	9.3 (6.5–13.6)	10.6 (6.5–15.3)	0.126
**Glucose (70–100 mg/dL)**	204.1 (110–372)	194.5 (101–367)	171.5 (69–365)	227.1 (92–373)	0.211
**Blood Urea Nitrogen (10–20 mg/dL)**	35 (15–54.9)	34 (15–57)	31 (15–51)	27 (16–46)	**0.031**
**Creatinine (<1 mg/dL)**	0.7 (0.5–1.1)	0.7 (0.5–1.1)	0.7 (0.5–1.1)	0.7 (0.5–1.1)	0.448
**C-reactive protein (<3 mg/L)**	9.3 (0.2–20)	6.5 (2–23.4)	9 (0.7–25)	13.4 (0.2–24)	0.137
**Total Cholesterol (<200 mg/dL)**	259 (107–359)	236 (107–302)	245 (108–356)	235 (137–359)	0.124
**HDL (40–60 mg/dL)**	30 (5–75)	33 (5–72)	33.5 (5–56)	37 (12–38)	0.354
**Triglyceride (<150 mg/dL)**	122.5 (60–258)	159.5 (59–298)	138 (60–321)	154 (72–299)	0.058
**Leukocyte (4.1–8.9 10^3^/uL)**	8.8 (5.3–15.2)	9 (4–15.1)	9 (4–15.2)	9.6 (3–15.3)	0.8
**Hemoglobin (12.4–14.8 g/l)**	14.2 (11.7–16.7)	14.5 (11.7–17.3)	14 (10.8–15.9)	15 (11.5–17.7)	0.16
**Platelet (15–450.000/mm^3^)**	168.5 (100–458)	182 (112–428)	177 (102–485)	182 (103–441)	0.87

BMI: Body mass index. SMMI index: Skeletal muscle mass index. HbA1c: Hemoglobin A1c. HDL: High-density lipoprotein cholesterol. LDL: Low-density lipoprotein cholesterol; *****
*Kruskal Wallis test.* Statistically significant in bold at *p* ≤ 0.05.

**Table 3 healthcare-11-00243-t003:** Review of muscle measurements and muscle test features of the patients according to NAFLD groups.

	Hand Dynamometer	Upper Arm Circumference	Calf Circumference	Get Up and Go Test
**Normal-Mild**	1.00	0.215	1.00	1.00
**Normal-Moderate**	**<0.001**	**<0.001**	**<0.001**	**0.032**
**Normal-Severe**	**<0.001**	**<0.001**	**<0.001**	0.242
**Mild-Moderate**	**0.003**	**<0.001**	**0.002**	**0.039**
**Mild-Severe**	**<0.001**	**<0.001**	**<0.001**	0.273
**Modarate-Severe**	0.333	0,977	0.108	1.00

Pairwise Post Hoc Evaluation. Mann–Whitney U test with Bonferroni correction was used. Statistically significant in bold at *p* ≤ 0.05.

**Table 4 healthcare-11-00243-t004:** NAFLD frequency in dynapenia-nondynapenia groups.

	Dynapenia Group (n,%)	Nondynapenia Group (n,%)
**Non-NAFLD**	4 (9.5%)	30 (32.3%)
**Mild NAFLD**	2 (4.8%)	32 (34.4%)
**Moderate NAFLD**	22 (52.4%)	26 (28.0%)
**Severe NAFLD**	14 (33.3%)	5 (5.4%)

BMI: Body mass index. SMMI index: Skeletal muscle mass index. HbA1c: Hemoglobin A1c. HDL: High-density lipoprotein cholesterol. LDL: Low-density lipoprotein cholesterol.

**Table 5 healthcare-11-00243-t005:** Review of all parameters according to dynapenia groups.

Med (Min-Max)	Dynapenia Group N = 42	Nondynapenia Group N = 93	*p*-Value
**Age (years)**	55 (45–65)	55 (45–65)	0.4771
**BMI (kg/m^2^)**	28.4 (17.4–40.7)	28.6 (17–49.3)	0.7472
**Muscle Strength (kg)**	24 (18–28)	40 (30–60)	**<0.001 ***
**Muscle Mass (kg)**	52.4 (31.1–61.3)	51.1 (29.3–70.5)	0.9642
**SMMI %**	40 (26.6–62)	39 (20.1–48.3)	0.1782
**Upper arm circumference (cm)**	20 (15–41)	26 (14–34)	**<0.001 ***
**Calf circumference (cm)**	30 (21–39)	32 (21–39)	**<0.001 ***
**Up and go test**	1 (1–3)	1 (1–4)	**0.0442 ***
**Hba1c (4.7–5.6%)**	10.4 (6.5–15.3)	9.4 (6.5–14.5)	0.0862
**Glucose (70–100 mg/dL)**	183 (70–373)	190 (69–372)	0.8552
**Blood Urea Nitrogen (10–20 mg/dL)**	31 (15–51)	34 (15–57)	0.2722
**Creatinine (<1 mg/dL)**	0.7 (0.5–1.1)	0.7 (0.5–1.1)	0.6512
**C-reactive protein (<3 mg/L)**	9 (0.2–23.4)	9 (0.2–25)	0.8972
**Total Cholesterol (<200 mg/dL)**	240 (107–356)	246 (107–359)	0.7942
**HDL (40–60 mg/dL)**	31.5 (5–43)	33 (5–75)	0.9202
**Triglyceride (<150 mg/dL)**	138 (60–305)	146 (59–321)	0.8512
**Leukocyte (4.1–8.9 10^3^/uL)**	9.4 (3–15.3)	9 (4–15.2)	0.9002
**Hemoglobin (12.4–14.8 g/l)**	14.4 (10.58–17.7)	14.3 (11.5–17.3)	0.2212
**Platelet (15–450.000/mm^3^)**	183.5 (101–485)	174 (100–458)	0.8872

BMI: Body mass index. SMMI index: Skeletal muscle mass index. HbA1C: Hemoglobin A1c. HDL: High-density lipoprotein cholesterol. LDL: Low-density lipoprotein cholesterol * Student’s *t*-test. Statistically significant in bold at *p* ≤ 0.05.

## Data Availability

The datasets generated during and/or analyzed during the current study are available from the corresponding author upon reasonable request.

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
