# Peer review of "The Association between Non-Alcoholic Fatty Liver Disease and Dynapenia in Men Diagnosed with Type 2 Diabetes Mellitus"

_healthcare, 2023, doi:10.3390/healthcare11020243_

Round 1
Reviewer 1 Report
In my opinion, the manuscript is generally interesting, nevertheless, I have the following comments:
- Abstract – In Results stays “There were significant differences in muscle strength, upper arm circumference, calf circumference, and up-and-go test scores between the NAFLD groups” – but it is not clear which groups are meant
- Introduction – lack of reference to the passage found in verses 50-52; lack of development of the SSMI abbreviation in verse 98
- Materials and Methods - Kruskal-Wallis test is used to compare more than two groups rather than two non-normally distributed variables as stated in the text.
- Results:
§ Table 2 - the title of the table does not explain much; misleading information on the statistical test applied (Kruskal Wallis test or Paired Samples t test?); no information on the multiple comparison test carried out and its results (in the Table or in the text below)
§ Verses 157-158 - should be removed
- Discussion:
§ The fragment “The mean HbA1c measured in the group with dynapenia was 10.4% and 9.4% in the group without dynapenia; However, the mean HbA1c was found to be higher in the dynapenia group, and no statistically significant difference was found between the two groups. Sarcopenia is more common in patients with lower BMI; however, it was reported that dynapenia is more common in patients with higher BMI, and it was reported that muscle disorders might be dynapenia in patients with T2DM rather than sarcopenia” (verses 210-217) is incomprehensible.
§ Figure 2 and results presented in should be better placed in the Results section.
In addition, missing from the introduction or discussion section is an explanation of the potential molecular mechanism of the relationship between dynapenia and NAFLD.
Author Response
Dear Reviewer,
We appreciate your suggestion regarding how to improve the value of our work. We have made adjustments in accordance with your suggestions. Please find attached the revised version.

Reviewer 2 Report
In this paper, the Authors explore the relationship between NAFLD and dynapenia severity in type II diabetes patients.
While there is indeed merit in the paper, I have some questions:
- Dynapenia, along with T2DM and NAFLD, is not exclusive to male subjects: rather, systematic reviews and metanalyses highlight how female subjects tend to show earlier signs of muscle strength loss. Furthermore, I disagree with what the Authors state in lines 178-179, as handgrip reference values are available in literature. While I do understand that such values might not be available for the specific population being studied, I feel it does not justify such a restrictive inclusion criterion: on the other hand, it would be quite crucial to acquire baseline data in both sexes.
- The study is defined as "prospective", and yet if I'm not mistaken there were no repeated measures.
- The cohort's age range (45-65 years), while indeed significant for evaluating dynapenia at its onset according to currently available epidemiological data, fails to take into account old age patients (65+ years), namely the range in which the condition's effects are more pronounced. Why were such patients excluded from the study?
- Lines 84-87 are unclearly worded.
- Line 98: please state the model and maker of the BIA equipment used, to ensure data reproducibility. Furthermore, skeletal mass muscle index (SMMI) has not had its acronym declared in the text before usage
- In the "conclusions" sections, the Authors state that "[...] maintaining adequate muscle strength in middle-aged male patients with T2DM may be an effective strategy to reduce the risk of NAFLD independent from fasting glucose, HbA1c, and BMI." but it is to my understandings that there were no interventions in this study, and the participants' physical activity/training habits were not assessed, only the results of their up-and-go test. How do the Authors make such a statement?
Author Response

(The authors gave the same response as above.)

Round 2
Reviewer 2 Report
I would like to thank the Authors for their reply and thorough review of the manuscript, and consider myself satisfied by the answers provided.
I would like to suggest another round of proofreading to ameliorate the text's readability and correct a few minor spelling mistakes.
Author Response
Dear reviewer,
Once again, thank you for the assessment of the manuscript. All spelling and grammatical errors have been corrected throughout the text. To prevent an oversight, we integrated the Grammarly program with the controls.
Sincerely.